# Intercellular Bioimaging and Biodistribution of Gold Nanoparticle-Loaded Macrophages for Targeted Drug Delivery

**Sehwan Kim [1,†]** , **Sung Hun Kang [2,†]** , **Soo Hwan Byun [3]** , **Hye-Jin Kim [4]** , **In-Kyu Park [5]** , **Henry Hirschberg [6] and Seok Jin Hong [7,*]**

[1]  Department of Biomedical Engineering, Beckman Laser Institute Korea, College of Medicine, Dankook University, 119, Dandae-ro, Dongnam-gu, Cheonan, Chungnam 31116, Korea; paul.kim@dankook.ac.kr

[2]  Department of Biomedical Sciences, Hallym University College of Medicine, Chuncheon 24252, Korea; malice23@nate.com

[3]  Department of Oral & Maxillofacial Surgery, Dentistry, Hallym University College of Medicine, Anyang 14068, Korea; purheit@daum.net

[4]  Tomocube Inc., 155, Sinseong-ro, Yuseong-gu, Daejeon 34109, Korea; hjkim@tomocube.com

[5]  Department of Biomedical Sciences, Chonnam National University Medical School, Gwangju 61469, Korea; pik96@jnu.ac.kr

[6]  Beckman Laser Institute and Medical Clinic, University of California, Irvine 1002 Health Sciences Rd, Irvine, CA 92617, USA; hhirshb@uci.edu

[7]  Department of Otorhinolaryngology-Head and Neck Surgery, Hallym University College of Medicine, Dongtan Sacred Heart Hospital, 7, Keunjaebong-gil, Hwaseong-si, Gyeonggi-do 18450, Korea

*  Correspondence: enthsj@hanmail.net; Tel.: +82-31-8086-2670

†  These authors contributed equally in this study.

**Abstract:** In order to effectively apply nanoparticles to clinical use, macrophages have been used as vehicles to deliver genes, drugs or nanomaterials into tumors. In this study, the effectiveness of macrophage as a drug delivery system was validated by biodistribution imaging modalities at intercellular and ex vivo levels. We focused on biodistribution imaging, namely, the characterization of the gold nanoparticle-loaded macrophages using intracellular holotomography and target delivery efficiency analysis using ex vivo fluorescence imaging techniques. In more detail, gold nanoparticles (AuNPs) were prepared with trisodium citrate method and loaded into macrophage cells (RAW 264.7). First, AuNPs loading into macrophages was confirmed using the conventional ultraviolet-visible (UV-VIS) spectroscopy and inductively coupled plasma-mass spectrometry (ICP-MS). Then, the holotomographic imaging was employed to characterize the intracellular biodistribution of the AuNPs-loaded macrophages. The efficacy of target delivery of the well AuNPs uptake macrophages was studied in a mouse model, established via lipopolysaccharide (LPS)-induced inflammation. The fluorescent images and the ex vivo ICP-MS evaluated the delivery efficiency of the AuNPs-loaded macrophages. Results revealed that the holotomographic imaging techniques can be promising modalities to understand intracellular biodistribution and ex vivo fluorescence imaging can be useful to validate the target delivery efficacy of the AuNPs-loaded macrophages.

**Keywords:** gold nanoparticle; macrophage; biodistribution; holotomography; drug delivery System

---

## 1. Introduction

There are several advantages in using gold nanoparticles (AuNPs) for drug delivery, including high biocompatibility, ease of size control, as well as the efficient release of therapeutic agents triggered

by stimuli such as glutathione, pH, and light [1,2]. However, targeted delivery of the nanoparticles poses a challenge on therapeutic modality. In principle, nanoparticles may increase therapeutic efficacy with reduced undesired side effects. Yet, the application of nano-drug delivery system as therapeutic agents has several limitations, including insufficient accumulation at the target site, cell toxicity, lack of targeting capability, and rapid elimination in the lesion site.

To address these issues, several approaches have been employed to improve delivery efficacy, including biodegradable and biocompatible polymerization or the attachment of an antibody to a target (a specific organ or cell) [3]. Besides, bacterial cells have also been studied as medical therapeutics because of the high efficiency of swimming to the target site. This is due to the tracing behaviors of the bacterial cells depending on the aero, photography, pH, and heat. To enhance its mobility and power to help the bacteria approaches to specific target cells, electropolymerized microtubes with external guidelines or multi-layered polymer electrolyte particles loaded with magnetic material was used [4,5]. Although bacterial-based delivery systems have many potential in the field of medical therapeutics, they have major obstacles such as toxic and immune responses [6]. In this reason, recently, there have been a growing attention on immune cells as vehicles or carriers for the delivery of therapeutic agents in cancer therapy and diagnosis due to their ability to migrate through the extracellular matrix into the tumor-inflammation microenvironments [7,8]. Inflammation is recognized as a hallmark of cancers [9]. This is because inflammatory cells and cytokines located in the tumor site may contribute to tumor growth, progression, and immunosuppression [10].

Macrophages known as tumor-associated macrophages (TAMs) are abundant in tumor sites [11]. The innate ability of macrophages to migrate and accumulate within damaged, hypoxic, or inflammatory tumor regions, makes them an ideal target as potential carriers of therapeutic drugs and imaging agents [12,13]. For example, Anselmo et al. reported that macrophages can migrate to inflamed brain tissue of patients with Alzheimer's and Parkinson's disease [14]. Similarly, Brynskikh et al. loaded bone marrow-derived macrophages with antioxidant enzymes encapsulated in micelles and found that the loaded macrophages were better accumulated in the brain than free micelles, resulting in a neuroprotective effect [15]. These results indicate that macrophages play significant roles in damaged, hypoxic, or inflammatory tumor regions while providing the feasibility as nanomedicine carriers. The alveolar macrophages collected from mouse have been widely used to deliver agents targeting the injured lung tissues. Alveolar-originated macrophages loaded with nanomaterials trigger phagocytosis and migrate to target inflammatory or tumor sites that are induced by lipopolysaccharide (LPS) [16].

Taken together, these recent studies have provided interesting insights into the new therapeutic modalities using macrophages conjugated with nano particles targeting the inflammatory circuits in a tumor microenvironment [17]. Therefore, the main contributions of this study are to investigate the feasibility of using macrophages loaded with gold nanoparticles (AuNPs) as a drug delivery system. First, we closely examined AuNP-loaded macrophages at the intracellular level by using a holotomograph microscope. Holotomographic microscopy is operated by Quantitative Phase Imaging (QPI) technology. Quantitative Phase Imaging (QPI) is an imaging technique that observe and quantify the phase shift that occurs when light passes through transparent objects. Thanks to the QPI technique, not only we are able to investigate the dynamic 3-D morphology of cells in label-free and in real-time, but we can also visualize the differences in refractive index, so we can observe sub-cellular organelles of the cells without any label.

In this study, we conducted AuNP uptake assays using epifluorescence conjugation into AuNP, to confirm the sub-cellular localization of AuNP. By using HT-2H, together with holotomography and 3-D epifluorescence imaging, we were able to pinpoint the accurate 3-D localization of gold nanoparticles inside the cells. Figure 1d shows that AuNP was localized at peri-nuclear region of the cell and especially located at basal plane of the cell in 3-D lateral view. Furthermore, lipid droplet on the macrophages were visualized by holotomography without staining.

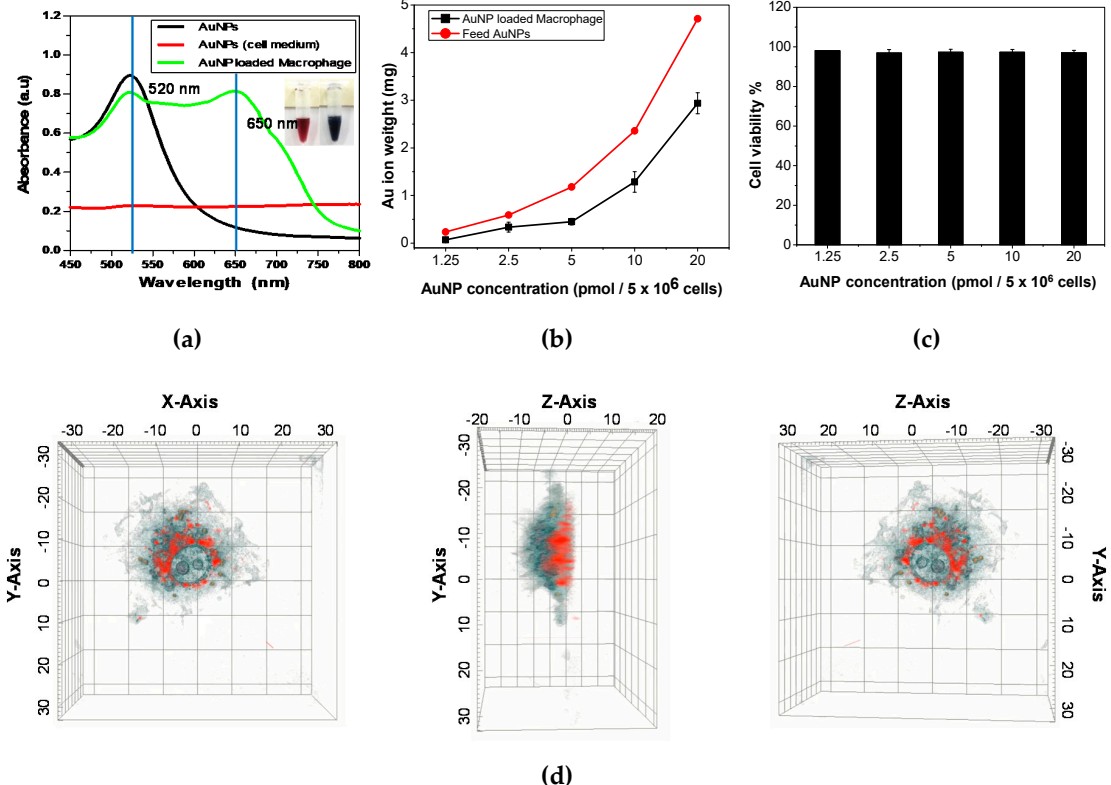

**Figure 1.** (**a**) Optical image and UV-visible spectrum of AuNPs, AuNPs (cell medium) and AuNP-loaded macrophages (RAW 264.7 cell line). The blue line represents the highest absorbance peak in the graph. AuNP(cell medium) spectra represented losing the characteristic peak of AuNPs in cell medium conditions (without cells). (**b**) Amount of AuNPs uptake in $5 \times 10^6$ macrophage cells determined by ICP-MS. (**c**) In vitro toxicity test of AuNP in the macrophage cell lines (RAW 264.7) for 24 h. (The cell viability test used $5 \times 10^6$ macrophages.) (**d**) 3D holotomographic image of AuNP-loaded macrophages; red indicates the locations of AuNPs, blue shows the cell membrane. The 3D image indicated cell membrane (blue) covers the AuNP (red) location in a three of direction view, which means that AuNPs (red) loaded well into macrophages. In the center of a cell, cell membrane color becomes dark due to the thickness of the cell membrane. (The grid lines show each plane of the x, y, and z axis and scale unit is um).

Then, we explored the delivery efficiency of the well AuNPs-loaded macrophages at ex vivo level. After labeling the macrophages with fluorescein-5-isothiocyanate (FITC), the accumulation level was analyzed by a fluorescence bioimaging technique. The biodistribution of the gold nanoparticles (AuNPs) were evaluated by inductively coupled plasma-mass spectrometry (ICP-MS) to determine the delivery efficiency of the AuNPs-loaded macrophages at the ex vivo level. The AuNPs-loaded macrophages confirmed by the holotomographic microscopy were resulted in the high target delivery efficacy at the ex vivo level. We believe that this is first study to investigate the ex vivo level biodistribution of AuNP-loaded macrophages by using a mouse model under the intravenous administration.

Therefore, in this study, the holotomographic imaging modality enables the gold nanoparticle-loaded macrophages to characterize at the intracellular level, as well as the fluorescence imaging techniques can evaluate the target delivery efficiency through the analysis of the ex vivo biodistribution.

## 2. Materials and Methods

### 2.1. Materials and Cell Line

We prepared gold (III) chloride hydrate and lipopolysaccharides (LPS; from Escherichia coli O55:B5) from Sigma-Aldrich (St. Louis, Missouri, USA), Trisodium citrate dehydrate from Junsei (Tokyo, Japan), Dulbecco's Modified Eagle's medium (DMEM), fetal bovine serum (FBS), penicillin 0.05%, MES buffer, EDC (1-ethyl-3-[3-dimethylaminopropyl]carbodiimide hydrochloride), and sulfo-NHS (N-hydroxysulfosuccinimide) from Thermo Fisher Scientific (Seoul, Korea), Lumiprobe BDP from BioCLONE (Seoul, Korea). Phosphate buffered saline (PBS) from BioNieer (Daejeon, Korea). Mouse macrophage cells (RAW 264.7; ATCC#TIB-71) were maintained in DMEM with high glucose concentration supplemented with 10% FBS and 100 U/mL penicillin 0.05%. All cultures were maintained at 37 °C in 10% $CO_2$.

### 2.2. Synthesis of AuNPs

AuNPs were prepared using the Turkevich method [18]. First, $HAuCl_4$ stock solution (10 mg/mL, 400 μL) was added to 20 mL of double distilled water heated at 97.5 °C and vigorously stirred for 30 min. Next, trisodium citrate hydrate (14.7 mg) was added to the $HAuCl_4$ solution and stirred for 30 min. Finally, the synthesis of AuNPs was confirmed by the appearance of red color in the solution. The solution was cooled down with stirring at room temperature and then stored at 4 °C for further use.

### 2.3. Macrophage Activation and the AuNP Loading Process

We used LPS for RAW 264.7 cell activation and morphology alteration from monocyte to macrophage. Macrophages were seeded in a 50 mL conical tube at a density of $5 \times 10^6$ cells per 2 mL of DMEM solution. Next, the LPS (2 μg) was added to the macrophage suspension solution. The cell suspension solution were incubated with AuNPs (1ml, 20 pmol, concentration calculated using UV-vis spectra method [19]) for 24 h at 37 °C in 5% CO2 and then rinsed three times with PBS prior to isolation by centrifugation at 1,400 rpm to remove excess non-ingested AuNPs. The AuNP-loaded macrophages were characterized using a holotomographic image (Tomocube HT-2H, Tomocube Inc., Daejeon, Korea), UV-visible spectrophotometer (OPTIZEN IV, Mecasys Co. Ltd, Korea), and inductively coupled plasma mass spectrometry (ICP-MS, 820-MS, Bruker, Germany).

### 2.4. Intracellular Biodistribution Analysis: in Vitro Experiment

The digital holotomographic imaging technique of Tomocube HT-2H (Tomocube Inc.) characterized the AuNP-loaded macrophages at the in vitro level by validating the degree of intracellular biodistribution between macrophages and AuNPs. Before analysis, macrophages (RAW264.7) were prepared in the microscopic dish (TomoDish, Tomocube Inc., Daejeon, Korea, with a #1.5H thickness and 50 mm diameter glass bottom) with LPS contained medium for activation. And AuNP solution was loaded into cell medium as following the "macrophage activation and the AuNP loading process" guide. In detail process of experiment, 3-D QPI and its correlative fluorescence images of live RAW 264.7 cells were obtained using a commercial holotomography (HT-2H, Tomocube Inc., Daejeon, Republic of Korea), which is based on Mach-Zehnder interferometry equipped with a digital micromirror device (DMD). A coherent monochromatic laser (λ = 532 nm) was divided into two paths, a reference and a sample beam, respectively, using a $2 \times 2$ single-mode fiber coupler. A 3-D refractive index (RI) tomogram was reconstructed from multiple 2-D holotomographic images acquired from 49 illumination conditions, a normal incidence, and 48 azimuthally symmetric directions with a polar angle (64.5°). The DMD was used to control the angle of an illumination beam impinging onto the sample [20]. The diffracted beams from the sample were collected using a high numerical aperture (NA) objective lens (NA = 1.2, UPLSAP 60XW, Olympus). The off-axis hologram was recorded by a CMOS image sensor (FL3-U3-13Y3MC, FLIR Systems). For 3-D epifluorescence imaging, AuNP-loaded macrophages were excited by a LED light source (575 nm). A total of 64 2-D sections within a 20-μm range were

acquired by moving the focus along the z-axis with a step size of 312 nm, immediately after acquiring a 3-D QPI image. Deconvolution of a reconstructed 3-D fluorescence images was performed by using commercial software (AutoQuant X3, Media Cybernetics). The visualization of 3-D RI maps and its correlative 3-D fluorescence signal with red pseudo-color was carried out using commercial software (TomoStudioTM, Tomocube Inc., Daejeon, Korea).

### 2.5. Ex vivo Biodistribution Analysis: Mouse Model Study

All animal experiments were performed under the guidelines of the Chonnam National University Medical School Research Institutional Animal Care Committee, and all the experimental protocols were approved by the committee. The macrophages were labeled with fluorescein-5-isothiocyanate (FITC 'isomer I', Invitrogen$^{TM}$, Eugene, Oregon, OR, USA) to determine the level of macrophage accumulation within each organ (brain, heart, liver, kidneys, lungs, and spleen). For the ex vivo imaging studies, we used 6–7-week-old Balb/c mice from Orient Bio INC (Seoul, Korea) which were administered LPS (1 mg/kg mice) intravenously and incubated for 24 h to induce lung inflammation.

A concentration of $1.0 \times 10^7$ cells/mouse of AuNP-loaded macrophages were prepared and intravenously injected to the inflammation mouse models. The animals were sacrificed at 6 h after injection (this experiment time was selected from results of static time-level data, and the organs such as brain, heart, liver, kidneys, lungs and spleen were collected. All the organs were washed with PBS and fixed using a 4% paraformaldehyde solution. Finally, the fluorescence images from each organ were analyzed using the Kodak imaging system (4000MN PRO, Kodak, USA) to confirm the biodistribution in the organs and to validate the target delivery efficiency at the specific site.

In addition, the biodistribution of the Au ions (from the AuNPs) that were loaded into macrophages was quantified using ICP-MS. The 6–7-week-old Balb/c mice (Orient Bio INC) were also prepared as the inflammation models using a 24 h treatment of LPS. The mice were intravenously administered $1.0 \times 10^7$ AuNP-loaded macrophage cells (containing $29.4 \pm 0.2$ µg of Au ions/mouse; N = 6). The animals were sacrificed, and the organs (brain, heart, liver, kidneys, lungs and spleen) were harvested at 1, 6, 12, 24, and 48 h post intravascular injection. The organs were washed with PBS and homogenized with PBS (0.3% Triton®X-100). Then, 5-fold aqua regia was added to each homogenized sample and sonicated for 2 h. The sonicated samples were then diluted to 100-fold with double distilled water and used for ICP-MS.

### 2.6. Statistical Analysis

Data are presented as mean ± standard deviation of results obtained from three independent trials unless otherwise indicated. Analysis of variance (ANOVA) (OriginPro8) was used to determine statistical significance between three or more groups, respectively. P-values < 0.05 were considered statistically significant.

## 3. Results

### 3.1. Characterization of AuNP-Loaded Macrophage

The AuNP-loaded macrophages (RAW 264.7) were characterized using the UV-visible spectrum to determine AuNP loading efficacy and the digital holotomography delineated the in vitro biodistribution through the reconstructed 3D holotomographic images. Apart from the characterization of the AuNP-loaded macrophages, in vitro toxicity test was also conducted.

Figure 1a delineates the absorption peak to show the characterization of AuNPs, AuNPs (cell medium) and AuNP-loaded macrophages. Due to the aggregation of the ingested AuNPs, the absorption peak was shifted from 520 nm to 650 nm. This shift, called as redshift, was measured by UV-Vis spectroscopy, which is a standard method to characterize AuNP-loaded macrophages on an absorbance spectrum [21]. Additionally, when AuNPs expose to cell medium, the aggregation is

induced, which provokes to lose the characteristic peak due to the high-level of the ionic component in the medium.

Figure 1b show the loading efficacy of AuNP in the macrophage over AuNP concentration. In this figure, the loading ratio of AuNP was increasingly dependent on the AuNP feed ratio and the highest loading efficacy of AuNPs is $61.80 \pm 9.26\%$ at 20 pmol/$5 \times 10^6$ cell conditions.

Figure 1c shows the toxicity of the AuNPs, which was verified by cell viability of macrophages. AuNP-loaded macrophages migrated to the disease site within 6 h, which is peak accumulation time of AuNPs in this inflammation model, and they were eliminated from the lung within 24 h. Based on this ground, we conducted the cell viability assay depending on various AuNP concentrations. The result confirmed that the AuNPs were not toxic because the viability of macrophages was more than 95% up to 20 pmol/$5 \times 10^6$ for 24 h.

MTT cell proliferation assay was used to determine the maximum loading concentration and subsequent toxicity of macrophage cells in Figure 1c. AuNP-loaded macrophages were able to retain over 95% cell viability until the maximum 20 pmol/$1 \times 10^6$ cells concentration was reached. Thus, we used the optimum concentration of 10 pmol/$1 \times 10^6$ cells for the remainder of this study.

Most of all, in this study, we described the ex vivo biodistribution for the AuNP-loaded macrophages using the digital holotomographic imaging technology. The in vitro biodistribution imaging technique enables the analysis of conjugation degree between AuNP nanoparticles and macrophages using 3D as well as tomographic images. Figure 1d reveals the distribution of AuNPs in the macrophages. The AuNPs are well uptake and distributed in the macrophages.

Furthermore, when AuNPs were absorbed into a macrophage, lipid droplets emerged inside the cell. In Figure 2a, the black dots indicate the lipid droplets in the macrophage cell. Moreover, the AuNPs were detected through fluorescence imaging technique. The macrophage cell is finally overlaid with lipid droplets which are indicated as black dots and the fluorescence image of the AuNPs are marked in red to characterize the in vitro biodistribution of the AuNP-loaded macrophage cell. Figure 2b is a time-lapsed image for the AuNP-loaded macrophages indicating cellular uptake and release of AuNPs in the macrophages during continual exposure to AuNP for 24 h. The emergence of lipid droplets explains the morphology activation of macrophage to phagocyte [22]. AuNP and lipid droplet detections confirm the activation and AuNP internalization of macrophages. The AuNP is easily detected by confocal microscopy with dye modification. However, lipid droplet staining induces phototoxicity and photobleaching under confocal microscopy analysis [23,24].

### 3.2. Target Site Biodistribution of AuNP-Loaded Macrophages

To validate the inflammation-homing ability of macrophages, the locations of RAW 264.7 cells distribution in healthy and inflammation-induced mice were identified through AuNP-loaded macrophage cells tracing in the body. FITC, a fluorescence dye, was used to label the cells and illumination from the stained cells determined the level of macrophage accumulation, as well as Au-ion intensity to indicate the concentration of AuNPs absorbed by macrophages in each organ.

Figure 3 shows the ex vivo biodistribution at the major target organs using the Au ion concentration per organ weight and the fluorescence imaging technique. The AuNP-loaded macrophages migrated to the LPS induced inflammation sites in the lungs of the mouse model up to 24 h. According to the results in Figure 3a, the delivery efficiency of AuNP increases with LPS and the number of macrophages. When the Ma-AuNP or Ma 1000 ($1000 \times 10^4$ ea-macrophage) cases, more AuNP were accumulated in the lungs of the LPS-induced inflammation than in the Ma-AuNP or Ma 500 (500ea-macrophages). Figure 3b shows the ex vivo biodistribution in each organ by fluorescence intensity, which confirms that macrophages and AuNPs move together without being separated in vivo. As a result, ex vivo fluorescence images clearly indicate that AuNP-loaded macrophage cells were delivered to the targeted organ; that is, the LPS-induced lung inflammation in this experiment.

In addition, the quantification analysis for Au ion accumulation was performed to analyze the static time level of the AuNP-loaded macrophages at each organ. After intravascular injection,

we extracted each organ at several time intervals to identify the accumulation changes of AuNP-loaded macrophages. Figure 4 describes the results that in the mice, the maximum accumulation of the Au ions were detected in the lungs at 6 hour but not in other organs (liver, heart, kidneys, spleen or brain). The accumulated AuNP-loaded macrophages in the lung were released within 24 h.

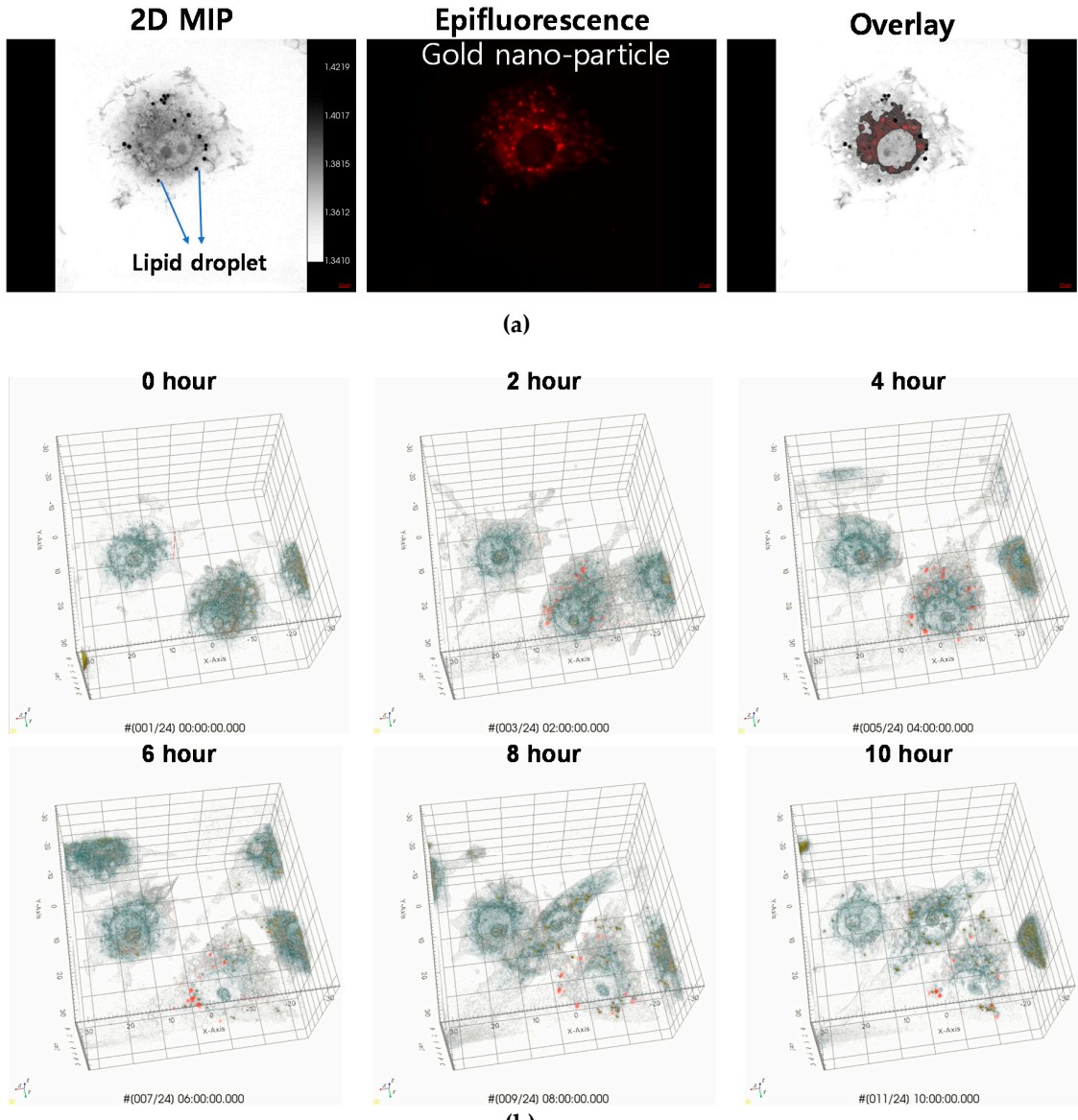

**Figure 2.** (**a**) The epifluorescence image of AuNP-loaded macrophage cells. Black dots and red epifluorescence indicate the lipid droplets and AuNPs, respectively, on a macrophage cell. (Lipid droplet RI value > 1.38). (**b**) Cellular uptake and release of AuNPs in macrophage cells. Cells were continuously exposed to AuNPs for 24 h, and an hourly time-lapse image was generated. After 5 h, the AuNP intensity showed the same result. (the grid lines show each plane of the x, y, and z axis and scale unit is um).

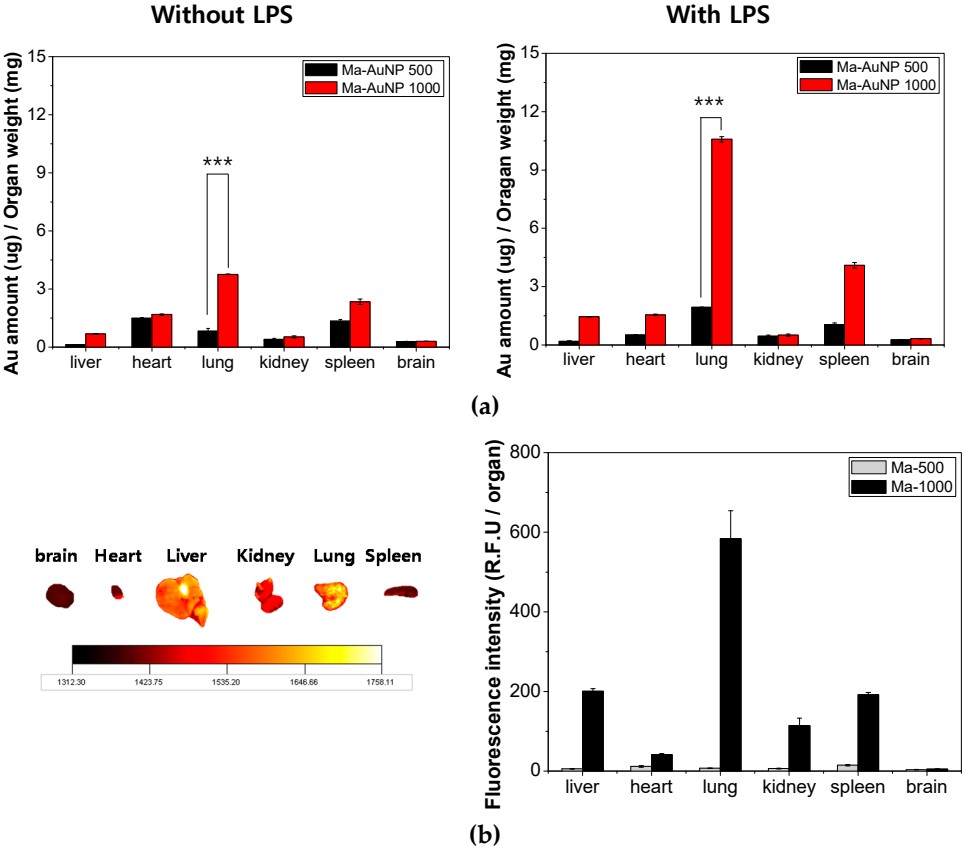

**Figure 3.** (**a**) Ex vivo Au ion quantification analysis in the major organs (brain, heart, liver, kidneys, lungs, and spleen) of mice with or without LPS-treatment. Au ions were extracted and detected from the AuNPs, which were loaded into the macrophages. (statistical analysis mark means *** $p < 0.05$) (**b**) Ex vivo fluorescence image and quantification analysis of the major organs (brain, heart, liver, kidneys, lungs, and spleen) at a 6 h post-injection. (determined by ICP-MS) (Ma-AuNP: AuNP loaded Macrophage, Ma: Macrophage, 500: $500 \times 10^4$ ea-macrophages, 1000: $1000 \times 10^4$ ea-macrophages) (the scale bare means the Mean of intensity).

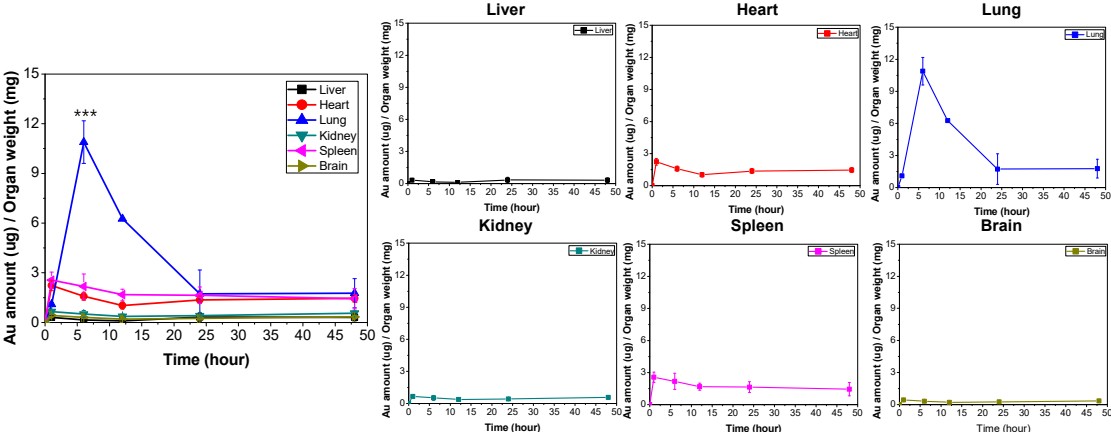

**Figure 4.** Quantification analysis of Au ion accumulation in the major organs (liver, heart, lungs, kidneys, spleen, and brain) obtained by ICP-MS. Each organ was collected at 1, 6, 12, 24, and 48 h after intravascular injection (N = 6). (statistical analysis mark means *** $p < 0.05$).

Therefore, the ex vivo fluorescence image and quantification analysis would be a potential bioimaging technique to validate the biodistribution of the AuNP-loaded macrophages in a nano-drug delivery system for cancer therapy.

## 4. Discussion

AuNPs are a potential candidate for biomedical applications such as bioimaging, diagnostics of nanomedicine, phototherapeutic agent, and targeted drug delivery system. They were traced to confirm their suitability as delivery vehicles for cell-based diagnostic platforms and nanotherapeutics. AuNPs exhibit ideal properties such as easy detection using ICP-MS, non-toxicity to cells, and ease of synthesis and conjugation with drugs.

Tumor and inflammation cells produce a wide spectrum of chemokines and growth factors that recruit circulating monocytes and differentiate them into macrophages [10,25]. The ability of macrophages to migrate and accumulate within tumor tissue, including hypoxic regions, makes them attractive vehicles for the delivery of diagnostic or therapeutic agents such as nanoparticles. For example, Taesoek Danieal Yang et al., reported the treatment and migration characteristics of AuShell-loaded macrophages at tumor site. They adopted intratumoral or intraperitoneal injection, which is a locally injected method into tumor or nearby, to compare the mobility characteristics of the AuShell-loaded macrophages at local tumor site [25].

On the contrary, our study employed AuNP-loaded macrophages (RAW 264.7) and injected them through the vein, called as intravenous injection, to emphasize on the target delivery ability of the AuNP-loaded macrophages in inflammation mouse model. In more detail, we prepared AuNP-loaded macrophage cells (RAW 264.7) and analyzed their intracellular distribution with a holotomographic imaging technique.

Holotomography can take microscopic images using QPI technology. This QPI technology is one of the useful tools for imaging live cells in non-invasive conditions. The basic theory of QPI is to measure the optical phase delay due to the difference in refractive index between the sample and the medium. No labeling agent is required because this tool collects the RI, the natural optical properties of the material. Especially, we adopted the QPI combined with fluorescence analysis which improves the spatiotemporal resolution with high molecular specificity. This synergistic called multimodal approaches [26].

Thanks to these advantages, differing from conventional microscope, holotomographic imaging modality is suitable to confirm the activation of macrophages because lipid droplets are highly feeble to light exposure and chemical fixation [23,24]. In fact, the emergence of lipid droplet is an evidence of the activated macrophages. When macrophages are activated with pro-inflammatory stimuli, lipids accumulate in a single membrane organelle that contains a core of neutral lipids, namely, triacylglycerol (TAG) and cholesterol ester [27]. Above all, the appearance of lipid droplets indicated that macrophages were well prepared by LPS-activation and ready for phagocytosis for AuNP uptake. In this sense, it could be crucial to see the lipid droplets on the macrophages. In this study, we succeeded to visualize these lipid droplets on macrophages by the holotomographic imaging in label-free fashion (Figures 1 and 2). AuNP-loading efficacy was determined by the conventional ICP-MS and the holotomographic imaging techniques revealed the in vitro intracellular distribution, confirming that AuNPs were successfully loaded into the macrophage cells by phagocytic activation. Macrophages incubated for 24 h with AuNPs presented extensive intracellular presence of the nanoparticles (Figures 1 and 2). This result suggests that macrophages can phagocytize AuNPs well and can be used as drug delivery carriers.

Several researchers have demonstrated that immune cells and nanoparticles can deliver therapeutic agents to the hypoxic tumor regions, using the innate phagocytic ability of the immune cells [13,28]. For targeted pharmacological therapy, drug delivery and release in target tissue sites have several limitations (such as toxicity, uncontrollable release, and accumulation) which are important criteria for consideration in nanomedicine research. The optimum physiological barrier of an endothelial

passage for macrophages is < 3 nm, and an increase in the thickness hampers the transfer of drugs to tissue sites. However, immune responses to infection, tissue injury, cancer, and inflammation are characterized by the extensive recruitment of immune cells. This response is induced by the rapid response to inflammation through activation and migration of macrophages to target tissue via intercellular route [29]. Choi et al. used the innate phagocytic capability of macrophages to deliver the therapeutic nanoparticles to tumor. These delivery vehicles transferred the nanoparticles into inaccessible tumor regions, such as a hypoxic area [30].

Our study scrutinized the distribution of AuNPs loaded macrophages in intracellular and ex vivo levels. In particular, we traced the biodistribution of that AuNP loaded macrophages to lung with high delivery efficacy. To the best of our knowledge, this is the first study that investigated the biodistribution of AuNP-loaded macrophages in vivo using a mouse model through intravenous administration. From our results, ex vivo fluorescent imaging and quantitative analyses of LPS-treated mice demonstrated that AuNP-loaded macrophage cells accumulate in the inflammatory lung sites within 6 h after intravenous injection (Figures 3 and 4). That is, it can indicate that AuNP-loaded macrophages crossed the endothelial barrier and migrated to the inflammation-induced tissue.

Numerous biological barriers exist to protect the human body from invasion by foreign particles. These barriers include the cellular and humoral arms of the immune system, such as mucosal barriers. A nanoparticle delivery system must overcome these barriers to reach the target tissue. Due to their specific migration and infiltration property, AuNP-loaded macrophages are well suited to target inflammatory tissue and overcome immune barriers in the liver, spleen, and kidneys. It is now well accepted that nano- and micro-vehicles are suitable for crossing the biological barriers through tissue diffusion, extravasation, and escape from hepatic filtration.

Recently, we also used another animal model toxicology study of intravenously injected AuNP-loaded macrophages to demonstrate that there are no tissue necrosis or infiltration of inflammatory corpuscles in the liver, lungs, and kidneys [31]. Thus, our findings support the use of macrophages as nanoparticle drug delivery vehicles in inflammation related diseases.

## 5. Conclusions

Nanoparticle-based delivery systems have been studied with various approaches to improve therapeutic and diagnostic efficacy. Among them, cell-based delivery approach can be a great strategy to overcome existing limitations such as poor accumulation, cell toxicity, and elimination by immune response. In particular, macrophage-based delivery vehicles offer several advantages compared to other cell-based delivery strategies thanks to easy to get, simple to load therapeutic agents, and free of re-injection into the bloodstream (e.g. immune response and toxicity) [32].

We prepared AuNP-loaded macrophages using RAW 264.7 cells and demonstrated their potential application as targeted delivery vehicles to inflammation sites using an LPS-treated mouse model.

At the intracellular level, the digital holotomographic imaging technique can characterize the AuNP-loaded macrophages by investigating the dynamic morphology of the cells and biodistribution imaging of the intracellular interaction between macrophages and AuNPs in label-free and in real-time. Besides, our ex vivo study of fluorescence imaging confirmed that well AuNPs uptake macrophages show the high delivery efficacy. Therefore, the digital holotomographic imaging technique would be a potential modality to understand intracellular biodistribution, while the ex vivo fluorescence imaging can be a promising technique to evaluate the target delivery efficiency through the analysis of the ex vivo biodistribution. Consequently, the AuNP-loaded macrophages can be a prospective candidate for targeting inflammatory regions. Furthermore, the use of nanotechnology mediated macrophages as tumor-promoting inflammatory targets may improve cancer therapies and further expand its diagnostic applications. Future research will confirm the application of nanomaterial-loaded immune cells in lung cancer models.

**Author Contributions:** S.K., S.H.K., H.H. and S.J.H. conceived and designed the study. S.H.K., H.-J.K., and S.J.H. collected the data and conducted the experiment. S.K., S.H.K., S.H.B., H.-J.K., I.-K.P., and S.J.H. analyzed and

interpreted the data. S.K., S.H.K., and S.J.H. drafted the manuscript for important intellectual content. S.H.B., I.-K.P., and H.H. involved in revising the manuscript it critically. All authors have read and agreed to the published version of the manuscript.

**Funding:** This research was supported by Leading Foreign Research Institute Recruitment & Young researcher Program through the National Research Foundation of Korea (NRF) funded by the Ministry of Science and ICT (MSIT) (NRF-2018K1A4A3A02060572, and NRF-2018R1C1B6008596). Also, this research was supported by Hallym University Research Fund.

**Conflicts of Interest:** The authors declare no conflict of interest.

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
