# Peer review of "Intercellular Bioimaging and Biodistribution of Gold Nanoparticle-Loaded Macrophages for Targeted Drug Delivery"

_electronics, doi:10.3390/electronics9071105_

Round 1
Reviewer 1 Report
Kindly see attached

Author Response
Bioimaging techniques for biodistribution analysis of gold nanoparticle-loaded macrophages for targeted drug delivery
Thank you for your all reviewers’ comprehensive comments and your valuable time to review. It is very helpful to improve our manuscript. All authors did our best to edit our original manuscript to follow the guidance from all reviewers’ comments. Again, we really appreciate it.
Reviewer’s comments #1
This is a well-developed paper with sound experimental methodology, results and analyses. Sufficient supporting references are also cited. The main weakness from this reviewer’s perspective is the lack of novelty and relevance of the work presented. To improve on the quality, it is recommended that the authors revise the paper to demonstrate what new information it provides beyond the ample of body of knowledge that already exists.
Q1-1) In the first place, the paper does not present any novel bioimaging techniques; holotomography and in particular, fluorescence imaging, are widely used for various biomedical imaging applications. The authors should thus justify why the need to present these imaging techniques for the purpose indicated.
R1-1) Thank you for your valuable suggestion. First of all, our holotomographic imaging can investigate the dynamic 3-D morphology of the cells in label-free and in real-time. We added the major characteristic of the holotomographic modality to justify the reasons why we need in the introduction section. In addition, in order to clarify our contributions, we have revised abstract, introduction, and conclusion.
Briefly speaking, main contributions are that
- Above all, the appearance of lipid droplets indicated that macrophages were well prepared by LPS-activation and ready for phagocytosis for AuNP uptake. In this sense, it could be crucial to see the lipid droplets on the macrophages. In this study, we succeeded to visualize these lipid droplets on macrophages by the holotomographic imaging in label-free fashion (Figure 1 and 2).
- To the best of our knowledge, this is the first study that investigated the biodistribution of AuNP-loaded macrophages in vivo using a mouse model through intravenous administration.
- Our results revealed that the holotomographic imaging techniques can be promising modalities to understand intracellular biodistribution and ex vivo fluorescence imaging can be useful to validate the target delivery efficacy of the AuNPs-loaded macrophages.
Abstract on pg 1) In order to effectively apply nanoparticles to clinical use, macrophages have been used as vehicles to deliver genes, drugs, or nanomaterials into tumors. In this study, the effectiveness of macrophage as a drug delivery system was validated by biodistribution imaging modalities at intercellular and ex vivo levels. We focused on biodistribution imaging, namely, the characterization of the gold nanoparticle-loaded macrophages using intracellular holotomography and target delivery efficiency analysis using ex vivo fluorescence imaging techniques. In more detail, gold nanoparticles (AuNPs) were prepared with trisodium citrate method and loaded into macrophage cells (RAW 264.7). First, AuNPs loading into macrophages was confirmed using the conventional ultraviolet-visible (UV-VIS) spectroscopy and inductively coupled plasma-mass spectrometry (ICP-MS). Then, the holotomographic imaging was employed to characterize the intracellular biodistribution of the AuNPs-loaded macrophages. The efficacy of target delivery of the well AuNPs uptake macrophages was studied in a mouse model, established via lipopolysaccharide (LPS)-induced inflammation. The fluorescent images and the ex vivo ICP-MS evaluated the delivery efficiency of the AuNPs-loaded macrophages. Results revealed that the holotomographic imaging techniques can be promising modalities to understand intracellular biodistribution and ex vivo fluorescence imaging can be useful to validate the target delivery efficacy of the AuNPs-loaded macrophages.
Introduction on pg 2-3) the main contributions of this study are to investigate the feasibility of using macrophages loaded with gold nanoparticles (AuNPs) as a drug delivery system. First, we closely examined AuNP-loaded macrophages at the intracellular level by using a holotomograph microscope. Holotomographic microscopy is operated by Quantitative Phase Imaging (QPI) technology. Quantitative Phase Imaging (QPI) is an imaging technique that observe and quantify the phase shift that occurs when light passes through transparent objects. Thanks to the QPI technique, not only we are able to investigate the dynamic 3-D morphology of cells in label-free and in real-time, but we can also visualize the differences in refractive index, so we can observe sub-cellular organelles of the cells without any label.
In this study, we conducted AuNP uptake assays using fluorescence conjugation into AuNP, to confirm the sub-cellular localization of AuNP. By using HT-2H, together with holotomography and 3-D epifluorescence imaging, we were able to pinpoint the accurate 3-D localization of gold nanoparticles inside the cells. Figure 1D shows that AuNP was localized at peri-nuclear region of the cell and especially located at basal plane of the cell in 3-D lateral view. Furthermore, lipid droplet on the macrophages were visualized by holotomography without staining.
Then, we explored the delivery efficiency of the well AuNPs-loaded macrophages at ex vivo level. After labeling the macrophages with fluorescein-5-isothiocyanate(FITC), the accumulation level was analyzed by a fluorescence bioimaging technique. The biodistirbution of the gold nanoparticles (AuNPs) were evaluated by inductively coupled plasma-mass spectrometry (ICP-MS) to determine the delivery efficiency of the AuNPs-loaded macrophages at the ex vivo level. The AuNPs-loaded macrophages confirmed by the holotomographic microscopy were resulted in the high target delivery efficacy at the ex vivo level. We believe that this is first study to investigate the ex vivo level biodistribution of AuNP-loaded macrophages by using a mouse model under the intravenous administration.
Therefore, in this study, the holotomographic imaging modality enables the gold nanoparticle-loaded macrophages to characterize at the intracellular level, as well as the fluorescence imaging techniques can evaluate the target delivery efficiency through the analysis of the ex vivo biodistribution.
Conclusions on pg10) Nanoparticle-based delivery systems have been studied with various approaches to improve therapeutic and diagnostic efficacy. Among them, cell-based delivery approach can be a great strategy to overcome existing limitations such as poor accumulation, cell toxicity, and elimination by immune response. In particular, macrophage-based delivery vehicles offer several advantages compared to other cell-based delivery strategies thanks to easy to get, simple to load therapeutic agents, and free of re-injection into the bloodstream (e.g. immune response and toxicity) [7,43]. We prepared AuNP-loaded macrophages using RAW 264.7 cells and demonstrated their potential application as targeted delivery vehicles to inflammation sites using an LPS-treated mouse model.
At the intracellular level, the digital holotomographic imaging technique can characterize the AuNP-loaded macrophages by investigating the dynamic morphology of the cells and biodistribution imaging of the intracellular interaction between macrophages and AuNPs in label-free and in real-time. Besides, our ex vivo study of fluorescence imaging confirmed that well AuNPs uptake macrophages show the high delivery efficacy. Therefore, the digital holotomographic imaging technique would be a potential modality to understand intracellular biodistribution, while the ex vivo fluorescence imaging can be a promising technique to evaluate the target delivery efficiency through the analysis of the ex vivo biodistribution. Consequently, the AuNP-loaded macrophages can be a prospective candidate for targeting inflammatory regions. Furthermore, the use of nanotechnology mediated macrophages as tumor-promoting inflammatory targets may improve cancer therapies and further expand its diagnostic applications. Future research will confirm the application of nanomaterial-loaded immune cells in lung cancer models.
Q1-2) Nanoparticle-loaded macrophages have been employed by several authors for various biomedical applications, including disease diagnosis and therapy- so why then demonstrate this in the paper for same applications? The non-toxicity of gold nanoparticles is also well known and hence no need to show this. This makes the results in Fig.1 and Fig.2 rather trivial. See, for example, the paper by Taeseok Daniel Yang et. al., “In vivo photothermal treatment by the peritumoral injection of macrophages loaded with gold nanoshells” (https://www.osapublishing.org/DirectPDFAccess/AA30509F-ADAA-FA63-4885276BC3B46FDC_333807/boe-7-1-185.pdf?da=1&id=333807&seq=0&mobile=no), where a similar biodistribution analysis is performed with fluorescence imaging on macrophages conjugated with Au nanoparticles.
R1-2) Thanks for your kind comments and recommending previous literature. As following your comment, we carefully investigated the Taesoek Daniel Yang et al.'s paper, titled “In vivo photothermal treatment by injection of gold nanoshell-loaded macrophages”. The Taesoek Daniel Yang et al.'s work employed the gold nanoshells for therapeutic agents, which is different nanomaterials with our AuNPs. This is the reason why we have analyzed the toxicology of our AuNP to the macrophages. Also, Taesoek Daniel Yang et al reported the migration property of the macrophages in the local tumor site by intratumoral or intraperitoneal injections. However, our AuNP-loaded macrophage used the intravenous injection for the check of the systemic organ distribution and target delivery properties (ex vivo experiment results figure 3 and 4). The revised manuscript has been updated based on the above investigation which is following as:
Discussion on pg8) Taesoek Danieal Yang et al., reported the treatment and migration characteristics of AuShell-loaded macrophages at tumor site. They adopted intratumoral or intraperitoneal injection, which is a locally injected method into tumor or nearby, to compare the mobility characteristics of the AuShell-loaded macrophages at local tumor site [37]. On the contrary, our study employed AuNP-loaded macrophages (RAW 264.7) and injected them through the vein, called as intravenous injection, to emphasize on the target delivery ability of the AuNP-loaded macrophages in inflammation mouse model. In more detail, we prepared AuNP-loaded macrophage cells (RAW 264.7) and analyzed their intracellular distribution with a holotomographic imaging technique.
Q1-3) Indicate the approval status of the animal study conducted in this paper from the relevant Institutional Animal Care and Use Committee (IACUC).
R1-3) Sorry for missing the IACUC approval status. In this revised manuscript, we added the approval status of IACUC. [IACUC No. CNU IACUC-H-2018-59]
Materials and method on pg 4) All animal experiments were performed under the guidelines of the Chonnam National University Medical School Research Institutional Animal Care Committee, and all the experimental protocols were approved by the committee.
Q1-4) In conclusion, the authors should clearly justify why the current scientific literature is not sufficient as to merit this paper’s publication.
R1-4) As mentioned in R1-1), in this revised manuscript, we have modified in conclusions to clearly explain our major contributions. In addition, we have modified further in abstract and introduction sections. We believe that this revision can clarify our scientific contributions.
Conclusions on pg10) Nanoparticle-based delivery systems have been studied with various approaches to improve therapeutic and diagnostic efficacy. Among them, cell-based delivery approach can be a great strategy to overcome existing limitations such as poor accumulation, cell toxicity, and elimination by immune response. In particular, macrophage-based delivery vehicles offer several advantages compared to other cell-based delivery strategies thanks to easy to get, simple to load therapeutic agents, and free of re-injection into the bloodstream (e.g. immune response and toxicity) [7,43]. We prepared AuNP-loaded macrophages using RAW 264.7 cells and demonstrated their potential application as targeted delivery vehicles to inflammation sites using an LPS-treated mouse model.
At the intracellular level, the digital holotomographic imaging technique can characterize the AuNP-loaded macrophages by investigating the dynamic morphology of the cells and biodistribution imaging of the intracellular interaction between macrophages and AuNPs in label-free and in real-time. Besides, our ex vivo study of fluorescence imaging confirmed that well AuNPs uptake macrophages show the high delivery efficacy. Therefore, the digital holotomographic imaging technique would be a potential modality to understand intracellular biodistribution, while the ex vivo fluorescence imaging can be a promising technique to evaluate the target delivery efficiency through the analysis of the ex vivo biodistribution. Consequently, the AuNP-loaded macrophages can be a prospective candidate for targeting inflammatory regions. Furthermore, the use of nanotechnology mediated macrophages as tumor-promoting inflammatory targets may improve cancer therapies and further expand its diagnostic applications. Future research will confirm the application of nanomaterial-loaded immune cells in lung cancer models.

Reviewer 2 Report
The authors study the suitability and effectiveness of macrophages as carriers to deliver gold nanoparticles to inflammation sites in a mouse model. The manuscript contains interesting and novel results but only a brief description of the experimental techniques. My impression is that, although the results are interesting, they do not fit quite into the scope and subject areas of the journal “Electronics”. This is even apparent from the listed keywords. I therefore recommend the authors to consider submission of the manuscript to a different journal or otherwise to explain in much more detail the experimental techniques, their novelty and relation to the journal’s subject areas.
In any case, the description is in parts ambiguous and lacks necessary clarity and should be revised, in particular the abstract and the title. Furthermore, the scope and main results needs to be clearly stated. In the present form title, abstract and conclusion emphasize rather different aspects of the study. The title emphasizes “bioimaging techniques”, the abstract the “effectiveness of macrophage as a drug delivery system” and the discussion the first “in vivo study on the biodistribution of AuNP-loaded macrophages". The subject should be described more coherently and in line with the scope of the journal.
Some comments to the text:
In the title and throughout the text the term “biodistribution of gold nanoparticle-loaded macrophages” is used. The authors should distinguish more clearly between the biodistribution of NPs inside the macrophages and the biodistribution of macrophages inside the mouse organs.
Line 26,27: In order to..: the logic of the sentence is not clear and should be clarified.
Line 28: “nanoparticle delivery system” is meant by “drug delivery system stated”; please clarify
Line 32: AuNPs loading..: Should read: The successful loading of AuNPs into..
Line 34&35 Do you mean the biodistribution of AuNPs inside the macrophages?
Line 62 The description of macrophages being reservoirs of pathogens is unclear here. Do you want to emphasize their “delivery capability”?
Line 82 ..uptake and release of AuNPs in the macrophage cells” Do you mean: ..cellular uptake of AuNPs into the macrophages and release out of the cells?
Line 111: Tomocube HT-2H is a commercial name and should be inside the brackets
Line 114 What are the SEM results?
Line 121 (3D) image
Line 150 The statistical analysis is unclear. What was statistically analyzed and where are the results described?
Figure 1B What does the red line in Figure 1 B (Feed AuNPs) show?
Figure 1C How was the cell viability measured? With how many cells?
Figure 1D What are the scales?
Line 173 Do you mean Figure 1(B)?
Line 174 Please clarify conflicting concentration values: 20 pmol/10^6 cells; in Figure 1 B: pmol/5x10^6 cells in Fig 1 C pmol/ml; in line 107 5x10^6cells per 2 ml
Line 180 and other text passages; I do not agree with the authors that this is an “in vivo” description or “in vivo” study. For an in vivo study the animal model needs to be alive during the measurements.
Line 183 “biodistribution of the AuNP-loaded macrophages”. This is unclear. Figure 1 D shows the distribution of AuNP inside a single (?) cell.
Line 185 What is the mentioned target side inside the cell?
Line 186 lipid droplets emerge inside the cell
Line 192 AuNP exposure time is unclear: 24 h or 6 h as indicated in line 204
Line 217, 281, Figure 3 a: “number of macrophages”, “1000ea”, “1000” and “500” in Figure 3 a should be clarified and explained.
Line 219 biodistribution in each organ
Figure 3A Add “determined by ICP-MS”.
Line 232 What is meant by “over a 24-h time line”?
Line 255 microscopy
Line 284, 285, 307 How is that an “in vivo” investigation?
Line 287 The AU amounts for some organs (heart, spleen) in Figure 3 and Figure 4 do not seem to be inline. Is there a difference between Fig 3 A and Fig 4? Are data in Figure 4 with or without LPS?
Author Response
Bioimaging techniques for biodistribution analysis of gold nanoparticle-loaded macrophages for targeted drug delivery
Thank you for your all reviewers’ comprehensive comments and your valuable time to review. It is very helpful to improve our manuscript. All authors did our best to edit our original manuscript to follow the guidance from all reviewers’ comments. Again, we really appreciate it.
Reviewer’s comments #2
Q2-1) The authors study the suitability and effectiveness of macrophages as carriers to deliver gold nanoparticles to inflammation sites in a mouse model. The manuscript contains interesting and novel results but only a brief description of the experimental techniques. My impression is that, although the results are interesting, they do not fit quite into the scope and subject areas of the journal “Electronics”. This is even apparent from the listed keywords. I therefore recommend the authors to consider submission of the manuscript to a different journal or otherwise to explain in much more detail the experimental techniques, their novelty and relation to the journal’s subject areas.
R2-1) Thank you for your kind concerns. in fact, we submitted this manuscript to the special issue, “Optical Sensing for biomedical Applications” which may cover our topic.
Q2-2) In any case, the description is in parts ambiguous and lacks necessary clarity and should be revised, in particular the abstract and the title. Furthermore, the scope and main results needs to be clearly stated. In the present form title, abstract and conclusion emphasize rather different aspects of the study. The title emphasizes “bioimaging techniques”, the abstract the “effectiveness of macrophage as a drug delivery system” and the discussion the first “in vivo study on the biodistribution of AuNP-loaded macrophages". The subject should be described more coherently and in line with the scope of the journal.
R2-2) Thank you for valuable pointing out and kind concerns. In order to address those issues, we have revised many parts such as title, abstract, introduction and conclusion to emphasize on our contributions in terms of intracellular-level holotomography (i.e. lipid droplet detection, the cellular uptake between AuNP and macrophages), ex vivo target delivery efficacy under the intravenous administration, the systemic organ distribution by fluorescence imaging and so on.
Title) Intercellular bioimaging and biodistribution of gold nanoparticle-loaded macrophages for targeted drug delivery
Abstract on pg 1) In order to effectively apply nanoparticles to clinical use, macrophages have been used as vehicles to deliver genes, drugs, or nanomaterials into tumors. In this study, the effectiveness of macrophage as a drug delivery system was validated by biodistribution imaging modalities at intercellular and ex vivo levels. We focused on biodistribution imaging, namely, the characterization of the gold nanoparticle-loaded macrophages using intracellular holotomography and target delivery efficiency analysis using ex vivo fluorescence imaging techniques. In more detail, gold nanoparticles (AuNPs) were prepared with trisodium citrate method and loaded into macrophage cells (RAW 264.7). First, AuNPs loading into macrophages was confirmed using the conventional ultraviolet-visible (UV-VIS) spectroscopy and inductively coupled plasma-mass spectrometry (ICP-MS). Then, the holotomographic imaging was employed to characterize the intracellular biodistribution of the AuNPs-loaded macrophages. The efficacy of target delivery of the well AuNPs uptake macrophages was studied in a mouse model, established via lipopolysaccharide (LPS)-induced inflammation. The fluorescent images and the ex vivo ICP-MS evaluated the delivery efficiency of the AuNPs-loaded macrophages. Results revealed that the holotomographic imaging techniques can be promising modalities to understand intracellular biodistribution and ex vivo fluorescence imaging can be useful to validate the target delivery efficacy of the AuNPs-loaded macrophages.
Introduction on pg 2-3) the main contributions of this study are to investigate the feasibility of using macrophages loaded with gold nanoparticles (AuNPs) as a drug delivery system. First, we closely examined AuNP-loaded macrophages at the intracellular level by using a holotomograph microscope. Holotomographic microscopy is operated by Quantitative Phase Imaging (QPI) technology. Quantitative Phase Imaging (QPI) is an imaging technique that observe and quantify the phase shift that occurs when light passes through transparent objects. Thanks to the QPI technique, not only we are able to investigate the dynamic 3-D morphology of cells in label-free and in real-time, but we can also visualize the differences in refractive index, so we can observe sub-cellular organelles of the cells without any label.
In this study, we conducted AuNP uptake assays using fluorescence conjugation into AuNP, to confirm the sub-cellular localization of AuNP. By using HT-2H, together with holotomography and 3-D epifluorescence imaging, we were able to pinpoint the accurate 3-D localization of gold nanoparticles inside the cells. Figure 1D shows that AuNP was localized at peri-nuclear region of the cell and especially located at basal plane of the cell in 3-D lateral view. Furthermore, lipid droplet on the macrophages were visualized by holotomography without staining.
Then, we explored the delivery efficiency of the well AuNPs-loaded macrophages at ex vivo level. After labeling the macrophages with fluorescein-5-isothiocyanate(FITC), the accumulation level was analyzed by a fluorescence bioimaging technique. The biodistirbution of the gold nanoparticles (AuNPs) were evaluated by inductively coupled plasma-mass spectrometry (ICP-MS) to determine the delivery efficiency of the AuNPs-loaded macrophages at the ex vivo level. The AuNPs-loaded macrophages confirmed by the holotomographic microscopy were resulted in the high target delivery efficacy at the ex vivo level. We believe that this is first study to investigate the ex vivo level biodistribution of AuNP-loaded macrophages by using a mouse model under the intravenous administration.
Therefore, in this study, the holotomographic imaging modality enables the gold nanoparticle-loaded macrophages to characterize at the intracellular level, as well as the fluorescence imaging techniques can evaluate the target delivery efficiency through the analysis of the ex vivo biodistribution.
Conclusions on pg10) Nanoparticle-based delivery systems have been studied with various approaches to improve therapeutic and diagnostic efficacy. Among them, cell-based delivery approach can be a great strategy to overcome existing limitations such as poor accumulation, cell toxicity, and elimination by immune response. In particular, macrophage-based delivery vehicles offer several advantages compared to other cell-based delivery strategies thanks to easy to get, simple to load therapeutic agents, and free of re-injection into the bloodstream (e.g. immune response and toxicity) [7,43]. We prepared AuNP-loaded macrophages using RAW 264.7 cells and demonstrated their potential application as targeted delivery vehicles to inflammation sites using an LPS-treated mouse model.
At the intracellular level, the digital holotomographic imaging technique can characterize the AuNP-loaded macrophages by investigating the dynamic morphology of the cells and biodistribution imaging of the intracellular interaction between macrophages and AuNPs in label-free and in real-time. Besides, our ex vivo study of fluorescence imaging confirmed that well AuNPs uptake macrophages show the high delivery efficacy. Therefore, the digital holotomographic imaging technique would be a potential modality to understand intracellular biodistribution, while the ex vivo fluorescence imaging can be a promising technique to evaluate the target delivery efficiency through the analysis of the ex vivo biodistribution. Consequently, the AuNP-loaded macrophages can be a prospective candidate for targeting inflammatory regions. Furthermore, the use of nanotechnology mediated macrophages as tumor-promoting inflammatory targets may improve cancer therapies and further expand its diagnostic applications. Future research will confirm the application of nanomaterial-loaded immune cells in lung cancer models.
Q2-3) Some comments to the text: In the title and throughout the text the term “biodistribution of gold nanoparticle-loaded macrophages” is used. The authors should distinguish more clearly between the biodistribution of NPs inside the macrophages and the biodistribution of macrophages inside the mouse organs.
R2-3) Thank you for your valuable suggestion. In accordance with your comment, we clearly explain to avoid the confusion between biodistribution NPs inside the macrophage and the biodistribution of macrophages inside the mouse organs in abstract and conclusion sections.
Abstract on pg 1) In order to effectively apply nanoparticles for clinical use, macrophages have been used as vehicles to deliver genes, drugs, or nanomaterials into tumors. In this study, the effectiveness of macrophage as a drug delivery system was validated by biodistribution imaging modalities at intercellular and ex vivo levels. We focused on biodistribution imaging, namely, the characterization of the gold nanoparticle-loaded macrophages using intracellular holotomography and target delivery efficiency analysis using ex vivo fluorescence imaging techniques. In more detail, gold nanoparticles (AuNPs) were prepared with trisodium citrate method and loaded into macrophage cells (RAW 264.7). First, AuNPs loading into macrophages was confirmed using the conventional ultraviolet-visible (UV-VIS) spectroscopy and inductively coupled plasma-mass spectrometry (ICP-MS). Then, the holotomographic imaging was employed to characterize the intracellular biodistribution of the AuNPs-loaded macrophages. The efficacy of target delivery of the well AuNPs uptake macrophages was studied in a mouse model, established via lipopolysaccharide (LPS)-induced inflammation. The fluorescent images and the ex vivo ICP-MS evaluated the delivery efficiency of the AuNPs-loaded macrophages. Results revealed that the holotomographic imaging techniques can be promising modalities to understand intracellular biodistribution and ex vivo fluorescence imaging can be useful to validate the target delivery efficacy of the AuNPs-loaded macrophages.
Conclusions on pg10) At the intracellular level, the digital holotomographic imaging technique can characterize the AuNP-loaded macrophages by investigating the dynamic morphology of the cells and biodistribution imaging of the intracellular interaction between macrophages and AuNPs in label-free and in real-time. Besides, our ex vivo study of fluorescence imaging confirmed that well AuNPs uptake macrophages show the high delivery efficacy. Therefore, the digital holotomographic imaging technique would be a potential modality to understand intracellular biodistribution, while the ex vivo fluorescence imaging can be a promising technique to evaluate the target delivery efficiency through the analysis of the ex vivo biodistribution. Consequently, the AuNP-loaded macrophages can be a prospective candidate for targeting inflammatory regions. Furthermore, the use of nanotechnology mediated macrophages as tumor-promoting inflammatory targets may improve cancer therapies and further expand its diagnostic applications. Future research will confirm the application of nanomaterial-loaded immune cells in lung cancer models.
Q2-4)
Line 26,27: In order to..: the logic of the sentence is not clear and should be clarified.
Line 28: “nanoparticle delivery system” is meant by “drug delivery system stated”; please clarify
Line 32: AuNPs loading..: Should read: The successful loading of AuNPs into..
Line 34&35 Do you mean the biodistribution of AuNPs inside the macrophages?
R2-4) Sorry for those confusions. In accordance with your comment, we have modified our manuscript in abstract of section.
Abstract on pg 1) In order to effectively apply nanoparticles to clinical use, macrophages have been used as vehicles to deliver genes, drugs, or nanomaterials into tumors. In this study, the effectiveness of macrophage as a drug delivery system was validated by biodistribution imaging modalities at intercellular and ex vivo levels. We focused on biodistribution imaging, namely, the characterization of the gold nanoparticle-loaded macrophages using intracellular holotomography and target delivery efficiency analysis using ex vivo fluorescence imaging techniques. In more detail, gold nanoparticles (AuNPs) were prepared with trisodium citrate method and loaded into macrophage cells (RAW 264.7). First, AuNPs loading into macrophages was confirmed using the conventional ultraviolet-visible (UV-VIS) spectroscopy and inductively coupled plasma-mass spectrometry (ICP-MS). Then, the holotomographic imaging was employed to characterize the intracellular biodistribution of the AuNPs-loaded macrophages. The efficacy of target delivery of the well AuNPs uptake macrophages was studied in a mouse model, established via lipopolysaccharide (LPS)-induced inflammation. The fluorescent images and the ex vivo ICP-MS evaluated the delivery efficiency of the AuNPs-loaded macrophages. Results revealed that the holotomographic imaging techniques can be promising modalities to understand intracellular biodistribution and ex vivo fluorescence imaging can be useful to validate the target delivery efficacy of the AuNPs-loaded macrophages.
Q2-5) Line 62 The description of macrophages being reservoirs of pathogens is unclear here. Do you want to emphasize their “delivery capability”?
R2-5) Thank you for your comment. Actually, we did not intend to do it. So, we have removed the mentioned sentence in this revised manuscript to avoid diversion of main topic.
(Page 2) Macrophages are immune cells that play various roles in human biology, especially show the migration property during an immune response. Macrophages known as tumor-associated macrophages (TAMs) are abundant in tumor sites
Q2-6) Line 82. uptake and release of AuNPs in the macrophage cells” Do you mean: ..cellular uptake of AuNPs into the macrophages and release out of the cells?
R2-6) Thank you for the thorough review and valuable comments. This could incur another confusion. In this revised manuscript, we have edited most related part of introduction session.
Introduction on pg 2) Taken together, these recent studies have provided interesting insights into the new therapeutic modalities using macrophages conjugated with nano particles targeting the inflammatory circuits in a tumor microenvironment [17]. Therefore, the main contributions of this study are to investigate the feasibility of using macrophages loaded with gold nanoparticles (AuNPs) as a drug delivery system. First, we closely examined AuNP-loaded macrophages at the intracellular level by using a holotomograph microscope. Holotomographic microscopy is operated by Quantitative Phase Imaging (QPI) technology. Quantitative Phase Imaging (QPI) is an imaging technique that observe and quantify the phase shift that occurs when light passes through transparent objects. Thanks to the QPI technique, not only we are able to investigate the dynamic 3-D morphology of cells in label-free and in real-time, but we can also visualize the differences in refractive index, so we can observe sub-cellular organelles of the cells without any label.
Q2-7)
Line 111: Tomocube HT-2H is a commercial name and should be inside the brackets
Line 114 What are the SEM results?
Line 121 (3D) image
R2-7) In accordance with your comment, we have corrected format and typos in materials and method section in this revised manuscript. Thank you for your pointing out.
Materials and method on pg 3) The AuNP-loaded macrophages were characterized using a holotomographic image (Tomocube HT-2H, Tomocube Inc., Daejeon, South Korea), UV-visible spectrophotometer (OPTIZEN IV, Mecasys Co. Ltd, South Korea), inductively coupled plasma mass spectrometry (ICP-MS, 820-MS, Bruker, Germany).
Q2-8) Line 150 The statistical analysis is unclear. What was statistically analyzed and where are the results described?
R2-8) Thank you for your great comment. we have used the statistical analysis in figure 3 and figure 4, respectively. Therefore, we marked the statistical analysis results as legends in these figures which are following as:
Figure 3. (A) Ex vivo Au ion quantification analysis in the major organs (brain, heart, liver, kidneys, lungs, and spleen) of mice with or without LPS-treatment. Au ions were extracted and detected from the AuNPs, which were loaded into the macrophages. (statistical analysis mark means *** <0.05) (B) Ex vivo fluorescence image and quantification analysis of the major organs (brain, heart, liver, kidneys, lungs, and spleen) at a 6 h post-injection. (determined by ICP-MS) (Ma-AuNP : AuNP loaded Macrophage, Ma: Macrophage, 500: 500 x 104 ea-macrophages, 1,000: 1,000 x 104 ea-macrophages) (the scale bare means the Mean of intensity)
Figure 4. Quantification analysis of Au ion accumulation in the major organs (liver, heart, lungs, kidneys, spleen, and brain) obtained by ICP-MS. Each organ was collected at 1, 6, 12, 24, and 48 h after intravascular injection (N = 6). (statistical analysis mark means *** <0.05)
Q2-9)
Figure 1B What does the red line in Figure 1 B (Feed AuNPs) show?
Figure 1C How was the cell viability measured? With how many cells?
Figure 1D What are the scales?
Line 173 Do you mean Figure 1(B)?
Line 174 Please clarify conflicting concentration values: 20 pmol/10^6 cells; in Figure 1 B: pmol/5x10^6 cells in Fig 1 C pmol/ml; in line 107 5x10^6cells per 2 ml
Line 180 and other text passages; I do not agree with the authors that this is an “in vivo” description or “in vivo” study. For an in vivo study the animal model needs to be alive during the measurements.
Line 183 “biodistribution of the AuNP-loaded macrophages”. This is unclear. Figure 1 D shows the distribution of AuNP inside a single (?) cell.
Line 185 What is the mentioned target side inside the cell?
Line 186 lipid droplets emerge inside the cell
R2-9) Thanks to your comments, we can recognize many typos and mistakes in our original manuscript. We scrutinized this revised manuscript to correct all these issues. All authors really appreciate your through review and kind comments. In accordance with your comment, we have revised our manuscript in results section and figures.
Figure 1. (A) Optical image and UV-visible spectrum of AuNPs, AuNPs (cell medium) and AuNP-loaded macrophages (RAW 264.7 cell line). The blue line represents the highest absorbance peak in the graph. AuNP(cell medium) spectra represented losing the characteristic peak of AuNPs in cell medium conditions (without cells). (B) Amount of AuNPs uptake in 5 × 106 macrophage cells determined by ICP-MS. (C) In vitro toxicity test of AuNP in the macrophage cell lines (RAW 264.7) for 24 h. (The cell viability test used 5x106 macrophages.) (D) 3D holotomographic image of AuNP-loaded macrophages. Red color indicates the locations of AuNPs. And Blue color shows the cell membrane. The 3D image indicated cell membrane (blue) covers the AuNP (red) location in a three of direction view, which means that AuNPs(red) loaded well into macrophages. In the center of a cell, cell membrane color becomes dark due to the thickness of the cell membrane. (The grid lines show each plane of the x, y, and z axis and scale unit is um)
Results on pg 6) Most of all, in this study, we described the ex vivo biodistribution for the AuNP-loaded macrophages using the digital holotomographic imaging technology. The in vitro biodistribution imaging technique enables the analysis of conjugation degree between AuNP nanoparticles and macrophages using 3D as well as tomographic images. Figure 1(D) reveals the distribution of AuNPs in the macrophages. The AuNPs are well uptake and distributed in the macrophages.
Furthermore, when AuNPs were absorbed into a macrophage, lipid droplets emerged inside the cell.
Q2-10) Line 192 AuNP exposure time is unclear: 24 h or 6 h as indicated in line 204
R2-10) In order to clarify it, we added more legends in Figure 2.
Figure 2. (A) The fluorescence image of AuNP-loaded macrophage cells. Black dots and red fluorescence indicate the lipid droplets and AuNPs, respectively, on a macrophage cell. (Lipid droplet RI value > 1.38). (B) Cellular uptake and release of AuNPs in macrophage cells. Cells were continuously exposed to AuNPs for 24 h, and an hourly time-lapse image was generated. After 5 hours, the AuNP intensity showed the same result. (Analysis time up to 24 hours) (the grid lines show each plane of the x, y, and z axis and scale unit is um)
Q2-11)
Line 217, 281, Figure 3 a: “number of macrophages”, “1000ea”, “1000” and “500” in Figure 3 a should be clarified and explained.
Line 219 biodistribution in each organ
Figure 3A Add “determined by ICP-MS”.
Line 232 What is meant by “over a 24-h time line”?
R2-11) Thank you for your valuable suggestion. In accordance with your comment, we have revised our manuscript in legends of figures and results section.
Figure 3. (A) Ex vivo Au ion quantification analysis in the major organs (brain, heart, liver, kidneys, lungs, and spleen) of mice with or without LPS-treatment. Au ions were extracted and detected from the AuNPs, which were loaded into the macrophages. (statistical analysis mark means *** <0.05) (B) Ex vivo fluorescence image and quantification analysis of the major organs (brain, heart, liver, kidneys, lungs, and spleen) at a 6 h post-injection. (determined by ICP-MS) (Ma-AuNP : AuNP loaded Macrophage, Ma: Macrophage, 500: 500 x 104 ea-macrophages, 1,000: 1,000 x 104 ea-macrophages) (the scale bare means the Mean of intensity)
Results on pg 7) According to the results in Figure 3(A), the delivery efficiency of AuNP increases with LPS and the number of macrophages. When the Ma-AuNP or Ma 1000 (1,000 x 104 ea-macrophage) cases, more AuNP were accumulated in the lungs of the LPS-induced inflammation than in the Ma-AuNP or Ma 500 (500ea-macrophages). Figure 3(B) shows the ex vivo biodistribution in each organ by fluorescence intensity, which confirms that macrophages and AuNPs move together without being separated in vivo.
Q2-12)
Line 255 microscopy
Line 284, 285, 307 How is that an “in vivo” investigation?
Line 287 The AU amounts for some organs (heart, spleen) in Figure 3 and Figure 4 do not seem to be inline. Is there a difference between Fig 3 A and Fig 4? Are data in Figure 4 with or without LPS?
R2-12) Thank you for your valuable suggestion. we have changed the “in vivo” to “ex vivo” and modified our manuscript in discussion and conclusion of section. We used different animal groups between Figures 3 and 4 to determine the comparison of the LPS-induced inflammation model with the normal group. Like your question, the heart and spleen show the gold ion accumulation in LPS-induced inflammation model.
This is because when injecting the LPS (lipopolysaccharide), it would be slightly exposed to the heart, liver, and spleen not only the lung. We thought that was the reason for spleen and other Au ion accumulation results. Though, still the LPS is a great option to induced inflammation, which like tumor-related inflammation stimulation in the lung site.
This revised manuscript has added more explanation to discussion session.
Discussion on pg 9) Our study scrutinized the distribution of AuNPs loaded macrophages in intracellular and ex vivo levels. In particular, we traced the biodistribution of that AuNP loaded macrophages to lung with high delivery efficacy. To the best of our knowledge, this is the first study that investigated the biodistribution of AuNP-loaded macrophages in vivo using a mouse model through intravenous administration. From our results, ex vivo fluorescent imaging and quantitative analyses of LPS-treated mice demonstrated that AuNP-loaded macrophage cells accumulate in the inflammatory lung sites within 6 h after intravenous injection (Figures 3 and 4). That is, it can indicate that AuNP-loaded macrophages crossed the endothelial barrier and migrated to the inflammation-induced tissue.

Reviewer 3 Report
My Review
The authors demonstrated the preparation of macrophage-based biohybrid drug delivery system and evaluated their potential use for biomedical applications by performing in vivo and ex vivo biodistribution analysis. Overall, the manuscript contains some original results and quite interesting findings. The topic of the manuscript is appropriate to the review for Electronics. However, there are several critical concerns, as indicated in detail below. Although a series of experiments are performed, the experimental design with some imaging techniques is rough and unclear to follow. Most importantly, the authors should consider revising the discussion section, which discuss the obtained results in depth. Considering that Electronics publishes considerably novel studies and findings in-depth understanding of materials, I would recommend the major revision of the manuscript. The intensive revision of the manuscript is required to improve the quality of the manuscript. Following suggestions and concerns should be clearly addressed.
In page 2, lines 54-61, the drug delivery system prepared in is one of the biohybrid systems, such as biohybrid microswimmers, microrobots, nanorobots using mammalian cells or bacteria. Therefore, the following studies should be introduced and cited in the introduction section to highlight the recent progress of cell-based active target drug delivery systems.
Multifunctional bacteria-driven microswimmers for targeted active drug delivery, ACS nano 11 (9), 8910-8923
Bioengineered and biohybrid bacteria-based systems for drug delivery, Advanced drug delivery reviews 106, 27-44
Biohybrid microtube swimmers driven by single captured bacteria, Small 13 (19), 1603679
In page 3, line 109, suggesting showing more details: how molar concentration of AuNPs is calculated; what the volume of AuNPs was added to what volume of solution containing the macrophages.
In page 3, lines 116-125, this section needs more details about the operational parameters of the Tomocube and some references for the technique.
In page 3, line 134, what is the reason for getting the mouse sacrificed at the specific time point? How 6 hr was chosen?
In page3, lines 162-163, the experiment for AuNP in cell culture medium (without cells) should be compared in Fig 1(a). The aggregation may be happened with the different ionic conditions and/or the presence of biomacromolecules. Moreover, I am not convinced that aggregation is directly correlated with the loading efficiency. They should be separately presented in the paragraph.
In page 4, lines 164-166, by looking at the Fig 1(c), it is not clear that the result confirms the AuNPs were “not toxic.” What if viability is decreased at 36 or 48 hours? The authors should elaborate why the 24 hour incubation matters for cell viability in this study.
In Fig 1(d), the titles of x-/y-axis are unclear and the units of them are missed. And explanation of colors is required.
In page 4, lines 177-179, Suggesting including additional experiments with higher concentrations to find out the maximum concentration of AuNPs, such as threshold. What if there is still no viability decreased at 30 or 40 pmol/ml?
In Fig 2, dimensions with scale bars are missed.
In Fig 3 (b), dimensions with scale bars are missed.
In the discussion section, suggesting removing the irrelevant parts and including more discussions in depth that are directly relevant to the obtained results. In lines 244-280, the paragraphs seem quite generous. It is hard to see the data in the result section are quantitatively or qualitatively discussed, particularly in these lines.
In page 8, line 281, please add some references for “many studies.”
In page 8, line 284 and 307, suggest omitting “in vivo.” As stated in the manuscript, the biodistribution with imaging techniques was conducted in in vitro and ex vivo.
In page 8, line 311, the authors should specify what tumor models will be studied with the developed target drug delivery system.
Author Response
Bioimaging techniques for biodistribution analysis of gold nanoparticle-loaded macrophages for targeted drug delivery
Thank you for your all reviewers’ comprehensive comments and your valuable time to review. It is very helpful to improve our manuscript. All authors did our best to edit our original manuscript to follow the guidance from all reviewers’ comments. Again, we really appreciate it.
Reviewer’s comments #3
The authors demonstrated the preparation of macrophage-based biohybrid drug delivery system and evaluated their potential use for biomedical applications by performing in vivo and ex vivo biodistribution analysis. Overall, the manuscript contains some original results and quite interesting findings. The topic of the manuscript is appropriate to the review for Electronics. However, there are several critical concerns, as indicated in detail below. Although a series of experiments are performed, the experimental design with some imaging techniques is rough and unclear to follow. Most importantly, the authors should consider revising the discussion section, which discuss the obtained results in depth. Considering that Electronics publishes considerably novel studies and findings in-depth understanding of materials, I would recommend the major revision of the manuscript. The intensive revision of the manuscript is required to improve the quality of the manuscript. Following suggestions and concerns should be clearly addressed.
Q3-1) In page 2, lines 54-61, the drug delivery system prepared in is one of the biohybrid systems, such as biohybrid microswimmers, microrobots, nanorobots using mammalian cells or bacteria. Therefore, the following studies should be introduced and cited in the introduction section to highlight the recent progress of cell-based active target drug delivery systems.
- Multifunctional bacteria-driven microswimmers for targeted active drug delivery, ACS nano 11 (9), 8910-8923
- Bioengineered and biohybrid bacteria-based systems for drug delivery, Advanced drug delivery reviews 106, 27-44
- Biohybrid microtube swimmers driven by single captured bacteria, Small 13 (19), 1603679
R3-1) Thanks for your kind comments and recommending good literatures. Following your comments, recent progress of cell-based active target drug delivery systems (recommended three papers) was included to the introduction section in this revised manuscript.
Introduction on pg 2) To address these issues, several approaches have been employed to improve delivery efficacy, including biodegradable and biocompatible polymerization or the attachment of an antibody to a target (a specific organ or cell) [3]. Besides, bacterial cells have also been studied as medical therapeutics because of the high efficiency of swimming to the target site. This is due to the tracing behaviors of the bacterial cells depending on the aero, photography, pH, and heat. To enhance its mobility and power to help the bacteria approaches to specific target cells, electropolymerized microtubes with external guidelines or multi-layered polymer electrolyte particles loaded with magnetic material was used [4,5]. Although bacterial-based delivery systems have many potential in the field of medical therapeutics, they have major obstacles such as toxic and immune responses [6].
Q3-2) In page 3, line 109, suggesting showing more details: how molar concentration of AuNPs is calculated; what the volume of AuNPs was added to what volume of solution containing the macrophages.
R3-2) Thank you for your valuable suggestion. In accordance with your comment, we added the detail method (e.g., volume of solution and calculated method of AuNPs) in materials and method section.
Materials and Methods on pg 3) The cell suspension solution were incubated with AuNPs (1ml, 20 pmol, concentration calculated using UV-vis spectra method[19]) for 24 h at 37 °C in 5% CO2 and then rinsed three times with PBS prior to isolation by centrifugation at 1,400 rpm to remove excess non-ingested AuNPs.
Q3-3) In page 3, lines 116-125, this section needs more details about the operational parameters of the Tomocube and some references for the technique.
R3-3) We have rewritten the detailed method and the operational parameter of holotomographic microscope in the materials and method section. Additionally, we added the operational parameters reference for explanation of the technique.
Materials and Methods on pg 3-4)
Intracellular biodistribution analysis: in vitro experiment
The digital holotomographic imaging technique of Tomocube HT-2H (Tomocube Inc.) characterized the AuNP-loaded macrophages at the in vitro level by validating the degree of intracellular biodistribution between macrophages and AuNPs. Before analysis, macrophages (RAW264.7) were prepared in the microscopic dish (TomoDish, Tomocube Inc. with a #1.5H thickness and 50 mm diameter glass bottom) with LPS contained medium for activation. And AuNP solution was loaded into cell medium as following the "macrophage activation and the AuNP loading process" guide. In detail process of experiment, 3-D QPI and its correlative fluorescence images of live RAW 264.7 cells were obtained using a commercial holotomography (HT-2H, Tomocube Inc., Daejeon, Republic of Korea), which is based on Mach-Zehnder interferometry equipped with a digital micromirror device (DMD). A coherent monochromatic laser (λ = 532 nm) was divided into two paths, a reference and a sample beam, respectively, using a 2 x 2 single-mode fiber coupler. A 3-D refractive index (RI) tomogram was reconstructed from multiple 2-D holotomographic images acquired from 49 illumination conditions, a normal incidence, and 48 azimuthally symmetric directions with a polar angle (64.5o). The DMD was used to control the angle of an illumination beam impinging onto the sample [20]. The diffracted beams from the sample were collected using a high numerical aperture (NA) objective lens (NA=1.2, UPLSAP 60XW, Olympus). The off-axis hologram was recorded by a CMOS image sensor (FL3-U3-13Y3MC, FLIR Systems). For 3-D epifluorescence imaging, AuNP-loaded macrophages were excited by a LED light source (575 nm). A total of 64 2-D sections within a 20-µm range were acquired by moving the focus along the z-axis with a step size of 312 nm, immediately after acquiring a 3-D QPI image. Deconvolution of a reconstructed 3-D fluorescence images was performed by using commercial software (AutoQuant X3, Media Cybernetics). The visualization of 3-D RI maps and its correlative 3-D fluorescence signal with red pseudo-color was carried out using commercial software (TomoStudioTM, Tomocube Inc.)
Q3-4) In page 3, line 134, what is the reason for getting the mouse sacrificed at the specific time point? How 6 hr was chosen?
R3-4) Thank you for your valuable suggestion. The collection time was selected from the result of static time-level data of AuNPs loaded macrophages (figure 4). This result shows the maximum accumulation intensity at 6 hour-post injection in the lung. That experiment time condition is great for the comparative study of normal and inflammation models. and we have modified the manuscript including this explanation in the materials and method, and results section.
Materials and Methods on pg 4) The animals were sacrificed at 6 h after injection (this experiment time was selected from results of static time-level data (Figure 4), and the organs such as brain, heart, liver, kidneys, lungs, and spleen were collected.
Results on pg 8) In addition, the quantification analysis for Au ion accumulation was performed to analyze the static time level of the AuNP-loaded macrophages at each organ. After intravascular injection, we extracted each organ at several time intervals to identify the accumulation changes of AuNP-loaded macrophages. Figure 4 describes the results that in the mice, the maximum accumulation of the Au ions were detected in the lungs at 6 hour but not in other organs (liver, heart, kidneys, spleen or brain). The accumulated AuNP-loaded macrophages in the lung were released within 24 hours.
Q3-5) In page3, lines 162-163, the experiment for AuNP in cell culture medium (without cells) should be compared in Fig 1(a). The aggregation may be happened with the different ionic conditions and/or the presence of biomacromolecules. Moreover, I am not convinced that aggregation is directly correlated with the loading efficiency. They should be separately presented in the paragraph.
R3-5) Thanks for your insightful comment. As your comments, we have repeated the experiment including the UV-vis spectrum analysis of AuNPs in the cell medium. In fact, the AuNPs lost the characteristic peak in the cell medium as shown in figure 1(A). This is because, high ionic condition (such as cell medium) induce the AuNP aggregation.
According to your comments and pointing out, the revised manuscript has included the explanation about correlation between aggregation data and cellular uptake in the result section.
Figure 1. (A) Optical image and UV-visible spectrum of AuNPs, AuNPs (cell medium) and AuNP-loaded macrophages (RAW 264.7 cell line). The blue line represents the highest absorbance peak in the graph. AuNP (cell medium) spectra represented losing the characteristic peak of AuNPs in cell medium conditions (without cells).
Results on pg 5) Figures 1(A) delineates the absorption peak to show the characterization of AuNPs, AuNPs (cell medium) and AuNP-loaded macrophages. Due to the aggregation of the ingested AuNPs, the absorption peak was shifted from 520nm to 650 nm. This shift, called as redshift, was measured by UV-Vis spectroscopy, which is a standard method to characterize AuNP-loaded macrophages on an absorbance spectrum [21]. Additionally, when AuNPs expose to cell medium, the aggregation is induced, which provokes to lose the characteristic peak due to the high-level of the ionic component in the medium.
Figures 1(B) show the loading efficacy of AuNP in the macrophage over AuNP concentration. In this figure, the loading ratio of AuNP was increasingly dependent on the AuNP feed ratio and the highest loading efficacy of AuNPs is 61.80 ± 9.26% at 20 pmol/5x106 cell conditions.
Q3-6) In page 4, lines 164-166, by looking at the Fig 1(c), it is not clear that the result confirms the AuNPs were “not toxic.” What if viability is decreased at 36 or 48 hours? The authors should elaborate why the 24 hour incubation matters for cell viability in this study.
R3-6) Our macrophage-based system focuses the target delivery efficiency within delivering time. Based on the ICP-MS study, it is required only 6 hour- delivery time, and then eliminated from the organ within 24 hours. This is the main reason why we conducted cell viability assay up to 24 hour incubation. In order to clarify this part, we revised the results session which is following as:
Results on pg 5) Figure 1(C) shows the toxicity of the AuNPs, which was verified by cell viability of macrophages. AuNP-loaded macrophages migrated to the disease site within 6 hours, which is peak accumulation time of AuNPs in this inflammation model, and they were eliminated from the lung within 24 hours. Based on this ground, we conducted the cell viability assay depending on various AuNP concentrations. The result confirmed that the AuNPs were not toxic because the viability of macrophages was more than 95% up to 20 pmol/5x106 for 24 hours.
Q3-7) In Fig 1(d), the titles of x-/y-axis are unclear and the units of them are missed. And explanation of colors is required.
R3-7) We have revised the figure 1(D) to clearly show the titles of x-/y-/z-axis and the units. Also, the explanation for colors has added to the legends of the figure 1 (D). In brief, red indicates the location of AuNPs; blue shows the cell membrane. Additionally, In the center of a cell, cell membrane color becomes dark due to the thickness of the cell membrane. The edited parts are following as:
Figure 1 on pg 5)
Figure 1. (A) Optical image and UV-visible spectrum of AuNPs, AuNPs (cell medium) and AuNP-loaded macrophages (RAW 264.7 cell line). The blue line represents the highest absorbance peak in the graph. AuNP(cell medium) spectra represented losing the characteristic peak of AuNPs in cell medium conditions (without cells). (B) Amount of AuNPs uptake in 5 × 106 macrophage cells determined by ICP-MS. (C) In vitro toxicity test of AuNP in the macrophage cell lines (RAW 264.7) for 24 h. (The cell viability test used 5x106 macrophages.) (D) 3D holotomographic image of AuNP-loaded macrophages; red indicates the locations of AuNPs, blue shows the cell membrane. The 3D image indicated cell membrane (blue) covers the AuNP (red) location in a three of direction view, which means that AuNPs(red) loaded well into macrophages. In the center of a cell, cell membrane color becomes dark due to the thickness of the cell membrane. (The grid lines show each plane of the x, y, and z axis and scale unit is um)
Q3-8) In page 4, lines 177-179, Suggesting including additional experiments with higher concentrations to find out the maximum concentration of AuNPs, such as threshold. What if there is still no viability decreased at 30 or 40 pmol/ml?
R3-8) Thanks for your insightful question. Based on your comment, we have been repeating the experiment to measure the amount of AuNPs uptake in the macrophage cells (5 x 106 cells) including higher concentration than 20 pmol/5 x 106 cells. The results added to Figure 1 and the AuNP uptake did not increase significantly when macrophage cells were incubated with AuNPs at 40 pmol/5 x 106 cells concentration. In addition, the loading efficacy was found to be highest at 20 pmol/5 x 106 cells concentrations (61.80 ± 9.26% at 20 pmol/5 x 106 cells and 40.00 ± 4.65 at 40 pmol/5 x 106 cells calculated from Figure 1b). This result showed that the optimal concentration of AuNP was 20 pmol / 5 x 106 cells. And we revised the manuscript in the section and the results in Figure 1.
Results on pg 5) Figures 1(B) show the loading efficacy of AuNP in the macrophage over AuNP concentration. In this figure, the loading ratio of AuNP was increasingly dependent on the AuNP feed ratio and the highest loading efficacy of AuNPs is 61.80 ± 9.26% at 20 pmol/5x106 cell conditions.
Figure 1. (A) Optical image and UV-visible spectrum of AuNPs, AuNPs (cell medium) and AuNP-loaded macrophages (RAW 264.7 cell line). The blue line represents the highest absorbance peak in the graph. AuNP(cell medium) spectra represented losing the characteristic peak of AuNPs in cell medium conditions (without cells). (B) Amount of AuNPs uptake in 5 × 106 macrophage cells determined by ICP-MS. (C) In vitro toxicity test of AuNP in the macrophage cell lines (RAW 264.7) for 24 h. (The cell viability test used 5x106 macrophages.) (D) 3D holotomographic image of AuNP-loaded macrophages. Red color indicates the locations of AuNPs. And Blue color shows the cell membrane. The 3D image indicated cell membrane (blue) covers the AuNP (red) location in a three of direction view, which means that AuNPs(red) loaded well into macrophages. In the center of a cell, cell membrane color becomes dark due to the thickness of the cell membrane. (The grid lines show each plane of the x, y, and z axis and scale unit is um)
Q3-9) In Fig 2, dimensions with scale bars are missed.
R3-9) We have revised the legends in figure 2 in this revised manuscript.
Figure 2. (A) The epifluorescence image of AuNP-loaded macrophage cells. Black dots and red epifluorescence indicate the lipid droplets and AuNPs, respectively, on a macrophage cell. (Lipid droplet RI value > 1.38). (B) Cellular uptake and release of AuNPs in macrophage cells. Cells were continuously exposed to AuNPs for 24 h, and an hourly time-lapse image was generated. After 5 hours, the AuNP intensity showed the same result. (the grid lines show each plane of the x, y, and z axis and scale unit is um)
Q3-10) In Fig 3 (b), dimensions with scale bars are missed.
R3-10) We have revised the legends in figure 3 (B) in this revised manuscript.
Figure 3. (B) Ex vivo fluorescence image and quantification analysis of the major organs (brain, heart, liver, kidneys, lungs, and spleen) at a 6 h post-injection. (determined by ICP-MS) (Ma-AuNP : AuNP loaded Macrophage, Ma: Macrophage, 500: 500 x 104 ea-macrophages, 1,000: 1,000 x 104 ea-macrophages) (the scale bare means the Mean of intensity)
Q3-11) In the discussion section, suggesting removing the irrelevant parts and including more discussions in depth that are directly relevant to the obtained results. In lines 244-280, the paragraphs seem quite generous. It is hard to see the data in the result section are quantitatively or qualitatively discussed, particularly in these lines.
R3-11) Thank you for your valuable suggestion. In accordance with your comment, we remove the unnecessary parts and added the detailed explanation in the discussion section which is following as:
Discussion on pg 8-9) Tumor and inflammation cells produce a wide spectrum of chemokines and growth factors that recruit circulating monocytes and differentiate them into macrophages [10,25]. The ability of macrophages to migrate and accumulate within tumor tissue, including hypoxic regions, makes them attractive vehicles for the delivery of diagnostic or therapeutic agents such as nanoparticles. For example, Taesoek Danieal Yang et al., reported the treatment and migration characteristics of AuShell-loaded macrophages at tumor site. They adopted intratumoral or intraperitoneal injection, which is a locally injected method into tumor or nearby, to compare the mobility characteristics of the AuShell-loaded macrophages at local tumor site [37].
On the contrary, our study employed AuNP-loaded macrophages (RAW 264.7) and injected them through the vein, called as intravenous injection, to emphasize on the target delivery ability of the AuNP-loaded macrophages in inflammation mouse model. In more detail, we prepared AuNP-loaded macrophage cells (RAW 264.7) and analyzed their intracellular distribution with a holotomographic imaging technique.
Holotomography can take microscopic images using QPI technology. This QPI technology is one of the useful tools for imaging live cells in non-invasive conditions. The basic theory of QPI is to measure the optical phase delay due to the difference in refractive index between the sample and the medium. No labeling agent is required because this tool collects the RI, the natural optical properties of the material. Especially, we adopted the QPI combined with fluorescence analysis which improves the spatiotemporal resolution with high molecular specificity. This synergistic called multimodal approaches[26].
Thanks to these advantages, differing from conventional microscope, holotomographic imaging modality is suitable to confirm the activation of macrophages because lipid droplets are highly feeble to light exposure and chemical fixation [23,24]. In fact, the emergence of lipid droplet is an evidence of the activated macrophages. When macrophages are activated with pro-inflammatory stimuli, lipids accumulate in a single membrane organelle that contains a core of neutral lipids, namely, triacylglycerol (TAG) and cholesterol ester [27]. Above all, the appearance of lipid droplets indicated that macrophages were well prepared by LPS-activation and ready for phagocytosis for AuNP uptake. In this sense, it could be crucial to see the lipid droplets on the macrophages. In this study, we succeeded to visualize these lipid droplets on macrophages by the holotomographic imaging in label-free fashion (Figure 1 and 2). AuNP-loading efficacy was determined by the conventional ICP-MS and the holotomographic imaging techniques revealed the in vitro intracellular distribution, confirming that AuNPs were successfully loaded into the macrophage cells by phagocytic activation. Macrophages incubated for 24 h with AuNPs presented extensive intracellular presence of the nanoparticles (Figures 1 and 2). This result suggests that macrophages can phagocytize AuNPs well and can be used as drug delivery carriers.
Q3-12) In page 8, line 281, please add some references for “many studies.”
R3-12) Thanks for the comments. We are added the representative example instead of mentioning to “many studies” in discussion section.
Discussion on pg 8) Tumor and inflammation cells produce a wide spectrum of chemokines and growth factors that recruit circulating monocytes and differentiate them into macrophages [10,25]. The ability of macrophages to migrate and accumulate within tumor tissue, including hypoxic regions, makes them attractive vehicles for the delivery of diagnostic or therapeutic agents such as nanoparticles. For example, Taesoek Danieal Yang et al., reported the treatment and migration characteristics of AuShell-loaded macrophages at tumor site. They adopted intratumoral or intraperitoneal injection, which is a locally injected method into tumor or nearby, to compare the mobility characteristics of the AuShell-loaded macrophages at local tumor site [37].
Q3-13) In page 8, line 284 and 307, suggest omitting “in vivo.” As stated in the manuscript, the biodistribution with imaging techniques was conducted in in vitro and ex vivo.
R3-13) Sorry for this confusion. The revised manuscript has corrected typos and improved the consistence of terminology. In accordance with your comment, we also removed “in vivo” in Discussion section.
Disucssion on pg 9) Our study scrutinized the distribution of AuNPs loaded macrophages in intracellular and ex vivo levels. In particular, we traced the biodistribution of that AuNP loaded macrophages to lung with high delivery efficacy. To the best of our knowledge, this is the first study that investigated the biodistribution of AuNP-loaded macrophages in vivo using a mouse model through intravenous administration. From our results, ex vivo fluorescent imaging and quantitative analyses of LPS-treated mice demonstrated that AuNP-loaded macrophage cells accumulate in the inflammatory lung sites within 6 h after intravenous injection (Figures 3 and 4). That is, it can indicate that AuNP-loaded macrophages crossed the endothelial barrier and migrated to the inflammation-induced tissue.
Q3-14) In page 8, line 311, the authors should specify what tumor models will be studied with the developed target drug delivery system.
R3-14) Thank you for your great comments. We plan to study further at lung cancer model so, we have added the detail tumor models in conclusion section.
Conclusion on pg 10) Future research will confirm the application of nanomaterial-loaded immune cells in lung cancer models.

Round 2
Reviewer 2 Report
The authors have carefully addressed all points raised by me and the other reviewers. By this thorough revision, the manuscript has been greatly improved in quality and clarity. I have no further comments and recommend the publication of the manuscript.
Reviewer 3 Report
The authors’ response letter and the revised manuscript have addressed the points. Therefore, I am satisfied with the revision made on the manuscript and would like to recommend the revised manuscript for publication in Electronics.